# Biological N Fixation and N Transfer in an Intercropping System between Legumes and Organic Cherry Tomatoes in Succession to Green Corn



**Gabriela Cristina Salgado** [1,*], **Edmilson Jose Ambrosano** [2], **Fabrício Rossi** [3], **Ivani Pozar Otsuk** [2], **Gláucia Maria Bovi Ambrosano** [4], **Cesar Augusto Santana** [5], **Takashi Muraoka** [1] and **Paulo Cesar Ocheuze Trivelin** [1]

[1] Nuclear Energy Center (CENA), University of São Paulo (USP), Piracicaba 13416000, Brazil; muraoka@cena.usp.br (T.M.); pcotrive@cena.usp.br (P.C.O.T.)
[2] Southcentral Regional Center, São Paulo Agribusiness Technology Agency (APTA), Piracicaba 13416000, Brazil; ambrosano@apta.sp.gov.br (E.J.A.); ivani@iz.sp.gov.br (I.P.O.)
[3] Biosystems Engineering Department, Faculty of Animal Science and Food Engineering (FZEA), University of São Paulo (USP), Pirassununga 13635900, Brazil; fabricio.rossi@usp.br
[4] Department of Social Dentistry, Biostatistics, Piracicaba Dentistry College, University of Campinas, Piracicaba 13416000, Brazil; glaucia@fop.unicamp.br
[5] Crop Science Department, Luiz de Queiroz College of Agriculture (ESALQ), University of São Paulo (USP), Piracicaba 13416000, Brazil; cesar.santana.srpq@hotmail.com
* Correspondence: salgado.gc@gmail.com

**Abstract:** The aim of this study was to investigate the transfer of N from different legumes to cherry tomatoes in the intercropping system under residual straw of the previous green corn crop using the [15]N natural abundance method. We also investigated the temporal variation in nitrogen transfer to a cherry tomato, the biological nitrogen fixation (BNF) of legumes, and the N concentration of green corn cultivated in the intercrop succession. The experimental design was a complete randomized block with eight treatments and five replications, described as follows: two controls consisting of a monocrop of cherry tomato with or without residual straw, cherry tomato and jack bean, sun hemp, dwarf velvet bean, mung bean, and white lupine or cowpea bean in intercropping system. The BNF was responsible for more than half of the N accumulated in the legumes. The N of legumes was transferred to cherry tomato in similar quantities, and the leaves and fruits of cherry tomato received more N transfer than shoots. It was shown that N transfer increases with the growth/development of cherry tomatoes. The intercropping system with legumes did not affect the [15]N natural abundance of leaves and the aboveground biomass of green corn cultivated in succession.

**Keywords:** green manure; [15]N natural abundance; N concentration

## 1. Introduction

Biological nitrogen fixation (BNF) by legumes (rhizobium symbiosis) has received attention as a source of nitrogen (N) that can replace a portion of synthetic N fertilizer in a rotation or intercropping system. This method can be used in organic farming and is environmentally friendly [1]. According to [2], sugarcane (leaves + stalk) absorbed an equivalent of 11.5 kg ha$^{-1}$ of N of *Crotalaria juncea* planted before the sugarcane in the rotation system. The intercropping system between sun hemp and Guinea grass or switchgrass has shown legume to non-legume transfers of N of 43 and 51%, respectively [3].

Nitrogen (N) is an essential nutrient for plants and is required in large quantities. The N uptakes of tomatoes and green corn are approximately 137 kg ha$^{-1}$ [4,5] and 125 kg ha$^{-1}$ [6,7], respectively. Nitrogen can be a limiting factor for increased productivity in organic agriculture. Low N availability and other nutrients from organic fertilizers might justify the lower productivity of organic agriculture than conventional agriculture [8,9].

The availability of N from organic fertilizers such as manure is a consequence of the mineralization activity of the soil [8]. However, the rate of soil mineralization is highly variable because it depends on the temperature, humidity, aeration, type of soil, and source of N. Moreover, the mineralization rate must occur in sync with the culture demand for N such that N deficiency and losses of this nutrient do not occur through leaching, volatilization, or denitrification [10–13].

Therefore, intercropping the main crops with legumes can increase N use in organic agriculture. A portion of N from legumes can be transferred to non-legumes through decomposition and mineralization of legume residues and leaf leaching, the release of ammonium gas, exudation of compounds with N from root nodules, and transfers of interconnected roots via mycorrhiza [14–16]. In addition, the use of legumes maintains or increases the total N present in the soil, which can be made available for the crop in succession [2]. However, the quantities of N transfer between plants are variable, depending on the seasons [17], the species/variety of legume and non-legume plants [3,18,19], the presence of mycorrhizae in the soil [14], the type of intercropping system (such as the distance between legumes and non-legumes plants) [17,20], the use of organic or conventional agriculture [2,21,22], and the type of soil. Hence, because numerous variables can affect N transfer, it is essential to study the best arrangements in terms of the species of legumes and the density of sowing to increase the transfer of N between plants.

The hypotheses of this study are given as follows: (1) N transfer occurs from legumes to cherry tomato; (2) this transfer varies according to the species companion of the legume; (3) the BNF and amount of N input through BNF varies according to the legume species; (4) the N derived from the soil and legumes is sufficient to supply the N demand of green corn. Our aim was (a) to investigate the transfer of N from different legumes to cherry tomato in the intercropping system under residual straw of the previous green corn crop using the $^{15}$N natural abundance method, and (b) evaluate the BNF of each legume tested and (c) the N concentration and $^{15}$N natural abundance of green corn cultivated in succession. We also investigated the temporal variation (2011/2012) in N transfer to the cherry tomato and the BNF of legume.

## 2. Materials and Methods

### 2.1. Site Description and Experimental Design

The research was carried out in Southeast Brazil at an agroecological experimental station (altitude of 540 m, 22°43' S, 47°38' W) from 2010 to 2013. The experimental area was used for pasture for 60 years, then for research with lettuce and beet in 2006.

The experimental area is a Rhodic Kandiudox [23] (Table 2). The average temperature and rainfall are 24.3 °C and 172.4 mm in the summer and 18.8 °C and 44 mm in the winter, respectively (Figure 1).

**Table 1.** Physical and chemical characteristics in depths between 0 and 20 cm.

| | |
|---|---|
| Clay | 36% |
| Silt | 23% |
| Fine sand | 28% |
| Coarse sand | 13% |
| pH (CaCl$_2$) | 6.0 |
| BS | 98.4 mmol$_c$ dm$^{-3}$ |
| CEC | 92.4 mmol$_c$ dm$^{-3}$ |
| OM | 32 g kg$^{-1}$ |
| V | 74% |
| M | 0% |
| N | 1.8 g kg$^{-1}$ |

**Table 2.** Physical and chemical characteristics in depths between 0 and 20 cm.

| | |
|---|---|
| $\delta^{15}N$ ($\delta air$) | +9.44‰ |
| P | 27 ppm |
| K | 6.4 mmol$_c$ dm$^{-3}$ |
| Ca | 47 mmol$_c$ dm$^{-3}$ |
| Mg | 15 mmol$_c$ dm$^{-3}$ |
| H+Al | 24 mmol$_c$ dm$^{-3}$ |
| Al | 0 ppm |
| S | 7 ppm |
| B | 0.21 ppm |
| Cu | 5 ppm |
| Fe | 46 ppm |
| Mn | 46 ppm |
| Zn | 6.4 ppm |

BS—base saturation, CEC—cation exchange capacity, OM—organic matter, V—percentage of base saturation, m—aluminum saturation (m).

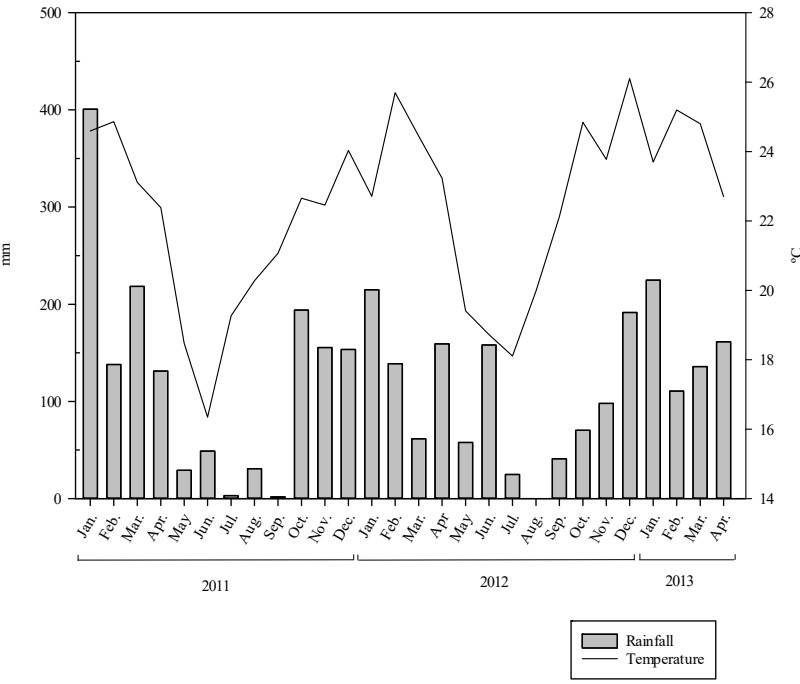

**Figure 1.** Temperature and rainfall during the months of the experiment, on average.

The experimental design was a complete randomized block with eight treatments and five replications, described as follows: (1) a control consisting of a monocrop of cherry tomato with residual straw from the previous green corn crop, (2) a control consisting of a monocrop of cherry tomato without residual straw from the previous green corn crop, (3) cherry tomato and jack bean (*Canavalia ensiformis* DC) in an intercropping system, (4) cherry tomato and sun hemp (*Crotalaria juncea* L.) in an intercropping system, (5) cherry tomato and dwarf velvet bean (*Mucuna deeringiana* (Bort)) in an intercropping system, (6) cherry tomato and mung bean (*Vigna radiata* (L.) Wilczek) in an intercropping system, (7) cherry tomato and white lupine (*Lupinus albus* L.) in an intercropping system, and (8) cherry tomato and cowpea bean (*Vigna unguiculata* (L.) Walp) in an intercropping system. All of

the intercropping systems were conducted with residual straw from the previous green corn crop.

## 2.2. Field and Crop Management

The experiment started with the sowing of green corn in January 2011 and intercropping tomato and legumes on straw corn residue in May 2011. This succession of green corn/tomato + legumes was repeated in 2012 with the sowing of green corn in January and intercropping of tomato and legumes in July. The experiment finished with the last green corn crop in January 2013 (Figure 2).

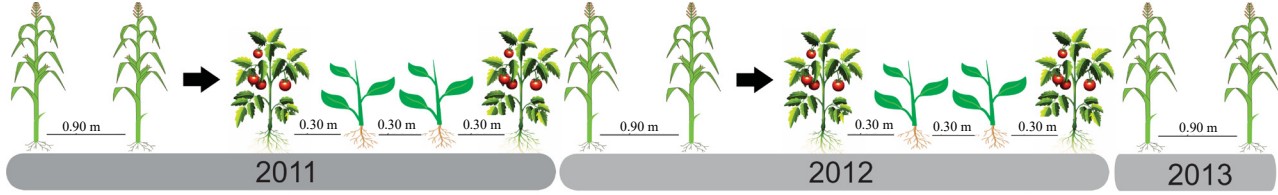

**Figure 2.** Illustration of succession system of green corn and cherry tomato/legumes with their respective spacing between the lines.

The area experimental was prepared with a moldboard plough and fertilizer with the following equivalent: (a) 25 Mg ha$^{-1}$ of organic manure with concentrations of 50 g kg$^{-1}$ OM, 1.5 g kg$^{-1}$ N, 0.80 g kg$^{-1}$ P$_2$O$_5$ (citric acid), 1.2 g kg$^{-1}$ K$_2$O, 3.1 g kg$^{-1}$ Ca, 0.6 g kg$^{-1}$ Mg and 0.6 g kg$^{-1}$ S; (b) 0.31 Mg ha$^{-1}$ thermophosphate with concentrations of 50 kg ha$^{-1}$ P$_2$O$_5$, 56 kg ha$^{-1}$ Ca, 21.85 kg ha$^{-1}$ Mg, 0.31 kg ha$^{-1}$ B, 0.16 kg ha$^{-1}$ Cu, 0.47 kg ha$^{-1}$ Mn, 31.20 kg ha$^{-1}$ Si, and 1.72 kg ha$^{-1}$ Zn; and (c) 0.1 Mg ha$^{-1}$ of potassium sulphate with concentrations of 50 kg ha$^{-1}$ K$_2$O and 15 kg ha$^{-1}$ S. Subsequently, green corn was sowed in January 2011 (Figure 2).

A Triton agricultural implement crushed the green corn plants after the green corn harvest. Therefore, the equivalents of 10 and 4 Mg ha$^{-1}$ of corn straw were on the ground in 2011 and 2012, respectively, for the subsequent no-till cherry tomato and legumes. The residual straw from the previous green corn crop was removed from the plot with single cherry tomato without straw. In 2012 and 2013, the green corn was not fertilized.

The cherry tomato and legumes were transplanted and sowed on the same day. The cherry tomato seed used was Access 21 from the Agronomy Institute (IA). This variety has good productivity and has been used by selected organic farmers [24]. The cherry tomato seedlings were produced in 128-cell expanded polystyrene trays in sprinkler-irrigated greenhouses. The seedlings were transplanted to the experimental area in pits (0.1 × 0.1 × 0.1 m) (Table 2). The pits were fertilized with 25 g of thermophosphate and 2.7 g of potassium sulphate, providing the equivalent of 4.4 g P$_2$O$_5$, 4.5 g Ca, 1.8 g Mg, 2.5 g Si, 10 mg Cu, 30 mg B, 80 mg Mn, 14 mg Zn, 1.4 g K$_2$O, and 0.5 g S. Ribbon was tied under two stems of the cherry tomatoes, and pruning to the eighth raceme was performed at 120 days. Each plot contained two rows of cherry tomatoes with six plants (Figure 2). The legumes were sowed in two lines between the cherry tomato rows (Figure 2). There was no nitrogen fertilization applied in the cherry tomato, and the source of N present in the area was the soil and green manure. The soil of the experiment has nitrogen bacteria fixation compatible with all legumes tested. Before this experiment, the other study and the pilot experiment were carried out in this soil, and nitrogen bacteria fixation was verified [19].

## 2.3. Sample Details

The green corn plant was sampled via the leaf (10 leaf plot$^{-1}$) in the R1 stage and the aboveground biomass without ears (10 plant plot$^{-1}$) in the R3 stage. The cherry tomato plant was sampled 20-shoot plot$^{-1}$ at 40 days after transplantation (DAT), and 20 leaf plot$^{-1}$ was sampled at 60 and 90 DAT in 2011 and 2012, respectively. The fruits were sampled at ten fruit plot$^{-1}$ in the mature stage at 150 DAT.

All aboveground biomass was sampled without pods at 40 and 100 days after seeding (DAS) for the legumes. The samples were separately dried in an oven at 65 °C with forced air circulation until they reached constant mass. Subsequently, the samples were ground in a Wiley mill and taken to the laboratory for analysis.

### 2.4. $^{15}N$ Natural Abundance ($\delta^{15}N$ ‰)

The samples were transported to the laboratory to determine the total nitrogen (%) and natural abundance of $^{15}N$ ($\delta^{15}N$ ‰). The analysis was performed using an isotope ratio mass spectrometer containing an automatic N analyzer connected to a mass spectrometer (IRMS)—N 20-20 ANCA GSL (Automatic Nitrogen and Carbon Analyzer, Gas, Solid and Liquid—SERCON) [25]. The standard formula expressed the natural abundance of $^{15}N$:

$$\delta^{15}N‰ = \left[ \left( \frac{R_{sample}}{R_{standard}} \right) - 1 \right] \times 10^3$$

The $R_{sample}$ and $R_{standard}$ are the ratios between $^{14}N/^{15}N$ of the sample and the atmosphere. This method is used for the calculation of biological nitrogen fixation (BNF) and N transfer. According to [26], this method is based on a slight $\delta^{15}N$ ‰ difference between N-fixation and other N sources.

### 2.5. Biological Nitrogen Fixation (BNF)

The percentage of N was derived from the atmosphere (%Ndfix) using the $^{15}N$ natural abundance method [26].

$$\%Ndfix = \frac{\delta^{15}N_{non-legume} - \delta^{15}N_{legume}}{\delta^{15}N_{non-legume} - \beta} \times 100$$

where $\delta^{15}N_{legume}$ is the aboveground biomass of legume plants, $\delta^{15}N_{non-legume}$ is the mean of the green corn leaf and the leaf of the cherry tomato in the monocrop that was growing in the same soil, and $\beta$ is the $\delta^{15}N$ of leaves of White lupine (−1.16‰) and Vigna (−1.48‰) grown hydroponically without soil [27,28]. The $\delta^{15}N$ of white lupine was used to calculate the %Ndfix of the sun hemp and white lupine, and the $\delta^{15}N$ of Vigna was used to calculate the %Ndfix of the other legumes tested in this research.

The accumulation of N was determined for the aboveground biomass of the legume using the N concentration and the dry mass of the plant fraction, expressed in kg ha$^{-1}$. The N accumulated by BNF percentage determined the nitrogen derived from biological nitrogen fixation (Ndf). Subsequently, the N derived from soil (Nds) was determined to subtract Ndf by the N accumulated in all aboveground biomass of legumes.

### 2.6. Nitrogen Transfer from Legume to Cherry Tomato

The calculation of N transfer from legume to cherry tomato was performed with the following formula:

$$\% N\ transfer = \left( 1 - \frac{\delta^{15}N\ ‰_{(m)}}{\delta^{15}N\ ‰_{(p)}} \right) \times 100$$

where %N transfer denotes the proportion of cherry tomato nitrogen derived from the legume, $\delta^{15}N$ ‰$_{(p)}$ is the monocrop of cherry tomato (reference plant), and $\delta^{15}N$ ‰$_{(m)}$ is the cherry tomato intercropped with the legume. On average, o $\delta^{15}N_{(p)}$ was +12.25‰ for the shoot and leaf and +13.54‰ for the fruit (Table 3). The reference plant was the leaf and the fruit of the same cherry tomato grown in the same soil in monocrop but one year before this experiment [22]. The N concentration was also determined by mass spectrometry analysis.

**Table 3.** Natural abundance $^{15}$N of leaf and fruit of reference plant of cherry tomato.

|  | Leaf | Fruit |
|---|---|---|
| **Repetitions** | $\delta^{15}N$ ‰ ($\delta$ar) | |
| 1 | +12.52 | +13.78 |
| 2 | +11.4 | +12.95 |
| 3 | +14.26 | +15.91 |
| 4 | +9.35 | +9.39 |
| 5 | +13.71 | +15.81 |
| Means | +12.25 | +13.57 |

Obs.: The table is taken from the work of Salgado et al. (2020).

*2.7. Statistical Analyses*

Statistical analysis with repeated measures was performed using the MIXED procedure in SAS software (Statistical Analysis System, 9.3). The Tukey–Kramer test was applied for comparisons between treatment means, and the F test was applied for comparisons between year means. The Dunnett test was used to contrast the effects of each treatment of cherry tomato with the additional treatment (reference plant) and each legume treatment with the additional treatment (non-legume plant) for $^{15}$N natural abundance ($\delta^{15}N$ ‰). The level of significance adopted for the analysis of variance was $p < 0.10$.

**3. Results**

*3.1. Biological Nitrogen Fixation*

There was a difference between legumes and non-legumes for the $^{15}$N natural abundance, except the dwarf velvet bean at 40 days after sowing (DAS) in 2011 and 2012 and white lupine 100 DAS in 2012 (Table 4). Therefore, the dwarf velvet bean at 40 DAS in 2011 and 2012 and the white lupine at 100 DAS in 2012 did not have biological nitrogen fixation (BFN) (Table 5). On average, the legumes $^{15}$N natural abundance varied between +0.49 and +5.20 in 2011 and +0.59 and +6.32 in 2012 (Table 4). In general, the smaller the $^{15}$N natural abundance, the higher the BNF.

**Table 4.** $^{15}$N natural abundance of leaves 40 days after sowing (DAS) and aerial part without grain 100 DAS of legumes in intercrop with cherry tomato.

| Treatments | 40 DAS [1] | | | 100 DAS [1] | | |
|---|---|---|---|---|---|---|
|  | **2011** | **2012** | **Average** | **2011** | **2012** | **Average** |
|  | $\delta^{15}N$ ‰ ($\delta$air) | | | | | |
| Jack bean | +5.20 aA * | +3.91 aA ** | +4.55 | +3.32 aA * | −1.17 bB ** | +1.08 |
| Sun hemp | +2.33 aBC ** | +1.24 aAB ** | +1.78 | +1.43 aA ** | +0.53 aB ** | +0.98 |
| Dwarf velvet bean | +5.83 aA ns | +5.55 aA ns | +5.69 | +0.49 aA ** | −0.63 aB ** | −0.07 |
| Mung bean | +4.36 aAB ** | +2.77 aAB ** | +3.56 | +2.06 aA ** | −0.59 aB ** | +0.73 |
| White lupine | +1.03 aC ** | +0.83 aB ** | +0.93 | +1.57 aA ** | +6.32 aA ns | +3.94 |
| Cowpea bean | +4.31 aAB ** | +1.74 aAB ** | +3.03 | +1.76 aA ** | +0.19 aB ** | +0.98 |
| Average | +3.84 | +2.67 | | +1.77 | +0.77 | |
| Non-legume | +8.43 | +8.43 | | +8.43 | +8.43 | |
| # CV (%) | 15.15 | | | 20.13 | | |

The means followed by the same lowercase letter in the lines do not differ by the F-test ($p < 0.10$), and means followed by the same uppercase letter in the columns do not differ by the Tukey–Kramer test ($p < 0.10$). [1] Statistics on transformed data for $\sqrt{(x)}$. * It differs from non-legume by the Dunnet test at 1%. ** It differs from non-legume by the Dunnet test at 5%. ns Not significant. # Coefficient of variation of data transformed.

**Table 5.** Biological nitrogen fixation of legumes at 40 days after sowing (DAS) and 100 DAS in intercrop with cherry tomato.

| Treatments | 40 DAS | | | 100 DAS | | |
|---|---|---|---|---|---|---|
| | **2011** | **2012** | **Average** | **2011** | **2012** | **Average** |
| | % | | | | | |
| Jack bean | 32.6 aC | 45.4 aB | 39.0 | 51.6 | 95.0 | 73.3 A |
| Sun hemp | 63.6 aAB | 75.2 aA | 69.4 | 73.0 | 82.6 | 77.8 A |
| Dwarf velvet bean | | | | 80.2 | 80.9 | 80.5 A |
| Mung bean | 41.0 aBC | 59.4 aAB | 50.2 | 64.4 | 85.2 | 74.8 A |
| White lupine | 77.0 aA | 79.2 aA | 78.1 | 79.3 * | | |
| Cowpea bean | 41.6 aBC | 67.6 aAB | 54.6 | 67.2 | 83.2 | 75.2 A |
| Average | 51.2 | 65.4 | | 67.3 b | 85.4 a | |
| [#] CV (%) | 24.01 | | | 20.39 | | |

The means followed by the same lowercase letter in the lines do not differ by the F-test ($p < 0.10$), and means followed by the same uppercase letter in the columns do not differ by the Tukey–Kramer test ($p < 0.10$). [#] Coefficient of variation of data transformed. * It does not include in the statistic because there was not BNF in the second year (2012)

　　　The sun hemp and white lupine showed the highest BNF, followed by cowpea bean, mung bean, and jack bean at 40 DAS in 2011 and 2012 (Table 5). However, there was no difference in BNF between legumes at 100 DAS, independent of the year (Table 5). The BFN was less in 2011 than in 2012 at 100 DAS, on average (Table 5). There was no difference of BNF ($p > 0.10$) between 40 and 100 DAS in 2011 for any legumes (Figure 3). In 2012, only the jack bean and mung bean showed a statistically increased BFN from 40 DAS to 100 DAS (Figure 3). However, in terms of numerical observation, the BFN in 100 DAS tended to be higher than that at 40 DAS (Figure 3).

　　　The highest total N accumulated of legumes was shown by jack bean followed by sun hemp, white lupine, cowpea bean, mung bean, and dwarf velvet bean at 100 DAS, on average (Table 6). A difference in N accumulated was observed between years only for dwarf velvet bean and cowpea bean (Table 6). The dwarf velvet bean had less total N accumulated in 2012 than in 2011. However, the seeds showed low emergence, not displaying all of the potential of this species. The cowpea bean showed more N accumulated in 2012 than in 2011 (Table 6). Consequently, in this same year, the cowpea bean had higher BNF (Table 5).

　　　The jack bean and cowpea bean showed a difference in the N accumulated derived from biological nitrogen fixation (Ndf) between years (Table 6). The jack bean and cowpea bean had higher Ndf in 2012 than in 2011. Nevertheless, only jack bean had lower N accumulated derived from soil (Nds) in 2012 than in 2011 (Table 6). The highest Ndf was shown by jack bean, followed by sun hemp, cowpea bean, dwarf velvet bean, and mung bean, on average (Table 6). Additionally, the highest Nds was found in white lupine, followed by jack bean, sun hemp, cowpea bean, dwarf velvet bean and mung bean, on average (Table 6).

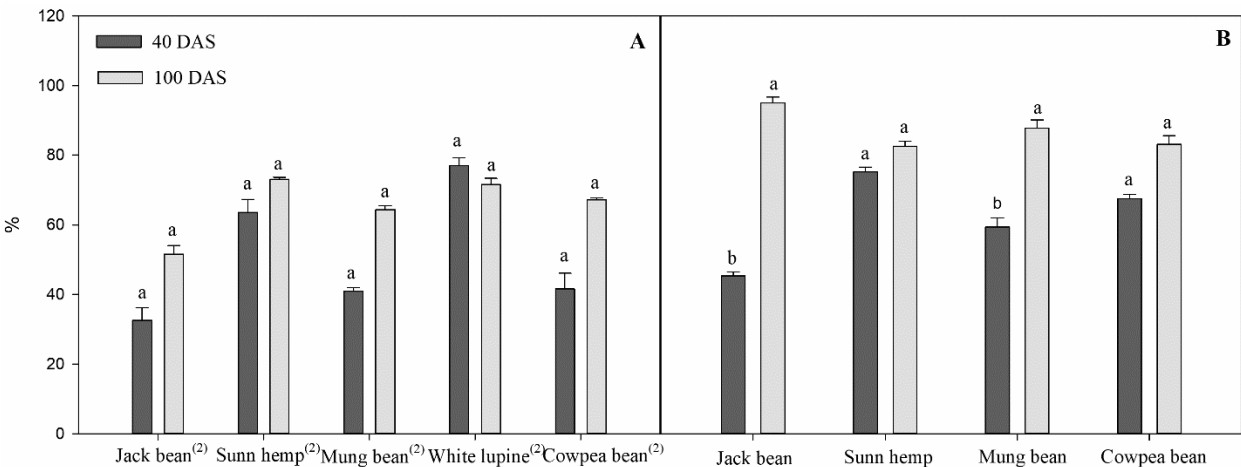

**Figure 3.** Biological nitrogen fixation (BNF) of legumes at 40 days after sowing (DAS) and 100 DAS in intercropping with cherry tomato in (**A**) 2011 and (**B**) 2012. The means followed by the same lowercase letter in the lines do not differ by the Tukey–Kramer test ($p < 0.10$). [2] Statistics on transformed data for log(x).

**Table 6.** Total accumulated nitrogen and the N accumulated derived from biological nitrogen fixation (Ndf) and nitrogen derived from soil (Nds) at 100 DAS.

| Treatments | N accumulated [1] | | | Ndf [1] | | | Nds [2] | | |
|---|---|---|---|---|---|---|---|---|---|
| | 2011 | 2012 | Average | 2011 | 2012 | Average | 2011 | 2012 | Average |
| | kg ha$^{-1}$ | | | | | | | | |
| Jack bean | 180.5 aA | 243.9 aA | 212.2 | 94.0 bAB | 233.9 aA | 164.0 | 86.6 aA | 10.0 bCD | 48.3 |
| Sun hemp | 186.7 aA | 160.6 aA | 173.6 | 135.7 aA | 128.6 aBC | 132.2 | 51.0 aAB | 31.9 aAB | 41.4 |
| Dwarf velvet bean | 89.2 aABC | 27.9 bC | 58.5 | 71.6 aAB | 27.2 aD | 49.4 | 17.6 aAB | 0.3 aD | 9.0 |
| Mung bean | 31.9 aC | 49.8 aBC | 40.9 | 21.0 aB | 45.6 aCD | 33.3 | 10.9 aB | 3.5 B aCD | 7.2 |
| White lupine | 106.4 aAB | 147.4 aAB | 126.9 | 66.6 * | | | 39.9 bAB | 147.4 aA | 93.6 |
| Cowpea bean | 64.0 bBC | 182.3 aA | 123.1 | 41.5 bB | 148.7 aB | 95.1 | 22.5 aAB | 33.6 aBC | 28.1 |
| Average | 109.8 | 135.3 | | 72.8 | 126.8 | | 38.1 | 37.8 | |
| # CV (%) | 25.22 | | | 24.21 | | | 40.41 | | |

The means followed by the same lowercase letter in the lines and means followed by the same uppercase letter in the columns do not differ by the Tukey–Kramer test ($p < 0.10$). [1] Statistics on transformed data for $\sqrt{}$(x). [2] Statistics on transformed data for log(x). # Coefficient of variation of data transformed. * It is not included in the statistics because there was no BNF in the second year (2012).

### 3.2. Cherry Tomato and Transfer of Nitrogen

No interactions were noted between treatment and year for the natural abundance of $^{15}$N, *% N transfer* from legumes to cherry tomato, and N concentration of shoots, leaves, and fruits of cherry tomato (Tables 7 and 8 and Figure 4).

**Table 7.** The natural abundance of $^{15}$N of shoots, leaves, and fruits of cherry tomato organic intercrop with legumes.

| Treatments | Shoots | | | Leaves | | | Fruits | | |
|---|---|---|---|---|---|---|---|---|---|
| | 2011 | 2012 | Average | 2011 | 2012 | Average | 2011 | 2012 | Average |
| | $\delta^{15}N$ ‰ ($\delta$ Air) | | | | | | | | |
| Control without straw | +8.82 ** | +9.68 * | +9.25 A | +8.13 ** | +7.07 ** | +7.60 A | +8.66 ** | +8.70 ** | +8.68 A |
| Control with straw | +8.45 ** | +8.99 ** | +8.72 A | +7.77 ** | +7.27 ** | +7.52 A | +8.59 ** | +9.71 ** | +9.15 A |
| Jack bean | +9.18 ** | +8.66 ** | +8.92 A | +7.20 ** | +6.58 ** | +6.89 A | +8.64 ** | +9.45 ** | +9.05 A |
| Sun hemp | +8.77 ** | +9.51 ** | +9.14 A | +6.66 ** | +6.8 ** | +6.78 A | +8.54 ** | +8.85 ** | +8.70 A |
| Dwarf velvet bean | +9.16 ** | +9.11 ** | +9.13 A | +6.69 ** | +7.04 ** | +6.87 A | +8.46 ** | +9.13 ** | +8.79 A |
| Mung bean | +8.34 ** | +9.26 ** | +8.80 A | +7.28 ** | +7.31 ** | +7.30 A | +8.32 ** | +8.69 ** | +8.50 A |
| White lupine | +8.91 ** | +8.70 ** | +8.81 A | +7.30 ** | +6.99 ** | +7.15 A | +8.23 ** | +9.74 ** | +8.98 A |
| Cowpea bean | +9.10 ** | +9.50 ** | +9.30 A | +6.98 ** | +6.98 ** | +6.98 A | +9.01 ** | +9.71 ** | +9.36 A |
| Average | +8.84 a | +9.18 a | | +7.25 a | +7.02 a | | +8.56 b | +9.25 a | |
| Reference plants | +12.25 | +12.25 | | +12.25 | +12.25 | | +13.57 | +13.57 | |
| # CV (%) | 12.02 | | | 13.32 | | | 11.74 | | |

The means followed by the same lowercase letter in the lines do not differ by the F-test ($p < 0.10$), and means followed by the same uppercase letter in the columns do not differ by the Tukey–Kramer test ($p < 0.10$). * Different from reference plant by the Dunnet test at 5%. ** Different from reference plant by the Dunnet test at 1%. # Coefficient of variation.

**Table 8.** N concentration of shoots, leaves, and fruits of cherry tomato intercrop with legumes in the field.

| Treatments | Shoots | | | Leaves | | | Fruits | | |
|---|---|---|---|---|---|---|---|---|---|
| | 2011 | 2012 | Average | 2011 | 2012 | Average | 2011 | 2012 | Average |
| | g kg$^{-1}$ | | | | | | | | |
| Control without straw | 38.3 | 34.0 | 36.2 A | 15.1 | 49.0 | 32.0 A | 24.4 | 20.6 | 22.5 A |
| Control with straw | 38.8 | 33.9 | 36.3 A | 14.8 | 46.9 | 30.8 A | 24.6 | 20.1 | 22.4 A |
| Jack bean | 36.0 | 32.4 | 34.2 A | 15.0 | 50.5 | 32.7 A | 24.1 | 20.3 | 22.2 A |
| Sun hemp | 37.8 | 29.9 | 33.9 A | 14.3 | 47.0 | 30.7 A | 23.0 | 20.8 | 21.9 A |
| Dwarf velvet bean | 35.1 | 34.7 | 34.9 A | 11.6 | 50.4 | 31.0 A | 25.0 | 21.7 | 23.3 A |
| Mung bean | 36.6 | 32.6 | 34.6 A | 14.8 | 48.8 | 31.8 A | 25.2 | 20.3 | 22.7 A |
| White lupine | 38.4 | 33.3 | 35.9 A | 16.4 | 49.4 | 32.9 A | 23.8 | 20.3 | 22.0 A |
| Cowpea bean | 35.2 | 34.0 | 34.6 A | 12.8 | 50.3 | 31.6 A | 24.1 | 19.2 | 21.7 A |
| Average | 37.0 a | 33.1 b | | 14.4 b | 49.0 a | | 24.2 a | 20.4 b | |
| # CV (%) | 13.64 | | | 13.44 | | | 8.19 | | |

The means followed by the same lowercase letter in the lines do not differ by the F-test ($p < 0.10$), and means followed by the same uppercase letter in the columns do not differ by the Tukey–Kramer test ($p < 0.10$). # Coefficient of variation of data transformed.

The natural abundance $^{15}$N showed a difference between the reference plant and each treatment of cherry tomato and legumes in the intercropping system ($p < 0.01$ or $0.05$) that had shown N transfer from legume to cherry tomato (Table 7). The smaller the $^{15}$N natural abundance of cherry tomato than the reference plant, the higher the N transfer from legume to cherry tomato. The natural abundance of $^{15}$N of the cherry tomato fruits in 2011 showed more depletion in $^{15}$N than in 2012 (Table 7). Therefore, more N transfer occurred from legumes to cherry tomato fruits in 2011 than in 2012, on average (Figure 4). Although the BNF was lower in 2011 than in 2012, the N transfer to cherry tomato was higher in 2011 than in 2012 (Table 6 and Figure 4).

Furthermore, the shoots had less N transfer than the leaves and fruits in both years (Figure 4), but the BNF showed no difference ($p < 0.1$) between 40 and 100 DAS in 2011 and 2012 (Figure 3). The fruits received less N transfer than the leaves in 2012, but the same did not occur in 2011 (Figure 4). Consequently, the N concentration in 2012 was lower than in 2011 (Table 8). However, the N concentration of leaves was higher in 2011 than in 2012 (Table 8).

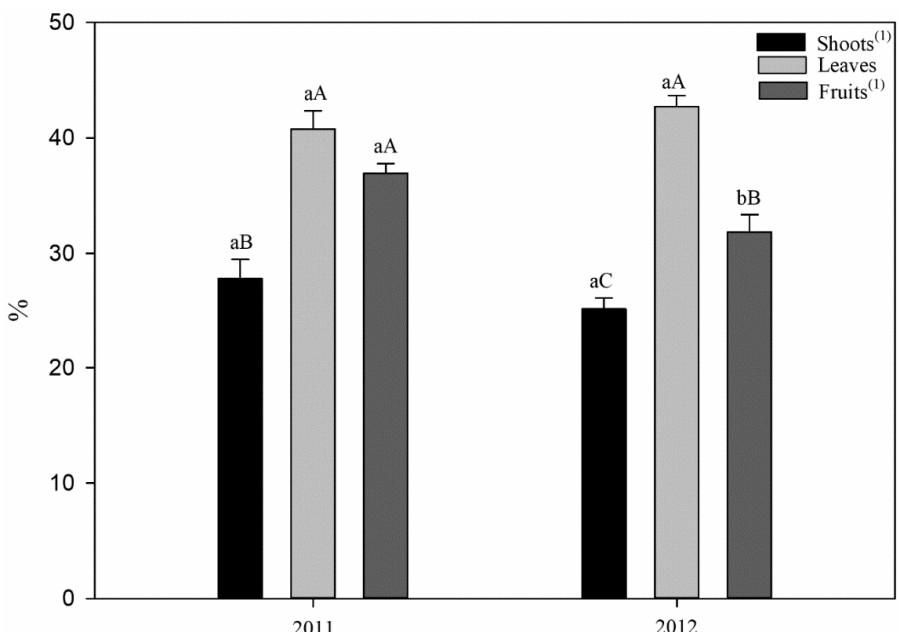

**Figure 4.** *% N transfer* from legumes to cherry tomato. The means followed by lowercase letters show differences between the years by the F-test ($p < 0.10$), and the means followed by uppercase letters show the difference between the parts of the plants (shoots, leaves, and fruit) by the Tukey–Kramer test ($p < 0.10$). [1] Statistics on transformed data for $\sqrt{(x)}$ for years.

### 3.3. Green Corn

There was no interaction between treatments and year for $^{15}$N natural abundance, N concentration, and the N accumulation of leaves of green corn (Table 9 and Figure 5). The N concentration of leaves was less in 2013 than in 2012 (Table 9). Consequently, N accumulation was also less in 2013 than in 2012 (Table 9). The aboveground biomass of green corn in the succession of dwarf velvet bean and cherry tomato in the intercropping system showed more N accumulation than the control with straw (monocrop of a cherry tomato), on average (Table 9). The $^{15}$N natural abundance of the leaves and aboveground biomass did not show a difference between treatments (Figure 5). The leaves in 2012 showed more depletion in $^{15}$N than in 2013 (Figure 5).

**Table 9.** N concentration of leaves and aerial part and N accumulated of aerial part of green corn.

| Treatments | Leaves | | | Aerial Part | | |
|---|---|---|---|---|---|---|
| | N concentration | | | N accumulated [1] | | |
| | 2012 | 2013 | Average | 2012 | 2013 | Average |
| | g kg$^{-1}$ | | | kg ha$^{-1}$ | | |
| Control without straw | 14.2 | 9.5 | 11.9 A | 15.25 | 11.93 | 13.60 AB |
| Control with straw | 15.5 | 9.9 | 12.7 A | 14.62 | 9.40 | 12.00 B |
| Jack bean | 14.1 | 11.1 | 12.6 A | 16.17 | 9.59 | 12.90 AB |
| Sun hemp | 15.6 | 10.9 | 13.3 A | 22.75 | 13.48 | 18.10 AB |
| Dwarf velvet bean | 15.9 | 9.6 | 12.7 A | 26.50 | 16.95 | 21.70 A |
| Mung bean | 15.6 | 8.9 | 12.2 A | 20.43 | 8.52 | 14.50 AB |
| White lupine | 15.6 | 11.0 | 13.3 A | 22.56 | 17.17 | 19.90 AB |
| Cowpea bean | 15.3 | 11.4 | 13.3 A | 20.28 | 5.67 | 13.00 AB |
| Average | 15.2 a | 10.3 b | | 19.82 a | 11.59 b | |
| # CV (%) | 13.90 | | | 22.07 | | |

The means followed by the same lowercase letter in the lines do not differ by the F-test ($p < 0.10$), and means followed by the same uppercase letter in the columns do not differ by the Tukey–Kramer test ($p < 0.10$). [1] Statistics on transformed data for $\sqrt{(x)}$. # Coefficient of variation of data transformed.

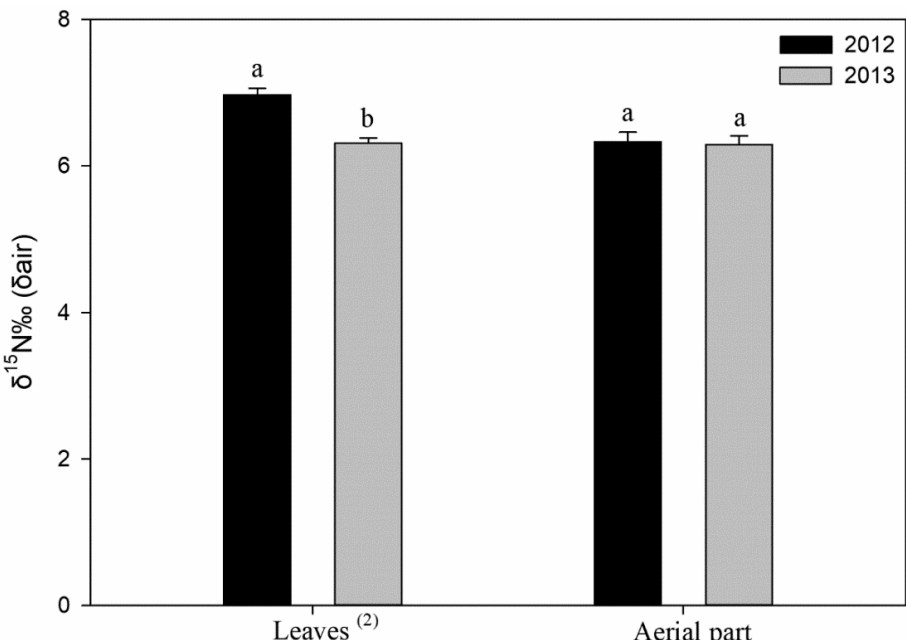

**Figure 5.** $^{15}$N natural abundance of leaves and aboveground biomass without ears of green corn. The means followed by the same lowercase letter do not differ by the F-test ($p < 0.10$). $^{(2)}$ Statistics on transformed data for log(x).

## 4. Discussion

Biological nitrogen fixation was responsible for more than half of N accumulated in the legumes (Tables 5 and 6), and the values were higher or similar to values in other studies [1,18,29,30]. However, the BNF was less in 2011 than in 2012 at 100 DAS (Table 5). This result could be due to a reduction of the N available in the soil. The utilization of corn straw (C: N ratio was 69:1) under the soil can cause the immobilization of the N [31–34], and the succession of the green corn without N fertilizer probably reduced the N available in the soil and forced the legumes to make an effort to increase BFN. Studies have shown a higher BFN of legumes under less fertilizer or without N fertilizer [21,35]. In chickpea and bitter vetch, authors in [36] verified that in soil with less N available, the N from BNF is higher than N from the soil.

The dry matter yield of legume was 17.9 Mg ha$^{-1}$ for white lupine, 12.5 Mg ha$^{-1}$ for sun hemp, 7.1 Mg ha$^{-1}$ for jack bean, 4.6 Mg ha$^{-1}$ for cowpea bean, 2.2 Mg ha$^{-1}$ for mung bean, and 2.4 Mg ha$^{-1}$ for dwarf velvet bean. Although jack bean did not have a higher dry matter yield, it accumulated higher Ndf (164 kg ha$^{-1}$) than the other legumes, mainly in 2012 (Table 6). The sun hemp (132.1 kg ha$^{-1}$) and cowpea bean (95.1 kg ha$^{-1}$) also accumulated a large quantify of N from BNF (Table 6). Authors in [1] reported average Ndf values for jack bean, velvet bean, and sun hemp of 210, 133, and 80 kg ha$^{-1}$, respectively. The BNF differs depending on species, edaphoclimatic condition, and agronomy management parameters such as temperature, rainfall, soil fertility, fertilizer, and sowing time [37–39].

In the present study, the transfer of N from legume to cherry tomato in the field (Figure 4) showed results similar to those obtained by [22] in a greenhouse. [40] verified approximately 15% of N transfer from broad bean to garlic. Authors in [41] demonstrated N transfer varying from 39 to 46% from the legume (*Pueraria phaseoloides*) to the rubber tree leaf. Moreover, another study with coffee and different legumes showed that 11.5 to 21.8 g kg$^{-1}$ of N in the leaves of coffee came from the N of legumes [18]. According to the same authors, *Cajanus cajan* and coffee in an intercropping system showed more N transferred to the coffee.

The N of legumes can be transferred to non-legume plants in an intercropping system as a portion of the N supply of these non-legume plants. The quantities of N transfer are

variable depending on the seasons [17], the species/variety of legume and non-legume plants [3,18,19], the presence of mycorrhizae in the soil [14], the type of intercropping system (such as the distance between legume and non-legume plants) [17,20], and fertilizer application [21].

The fruits and shoots of cherry tomato received less N transfer from legumes in 2012 than in 2011, although BNF was higher in 2012 (Table 5 and Figure 4). The same result occurred in the study by [22], in which cherry tomato leaf had less N transferred in the second year of cultivation in a greenhouse. The lower N transfer and N concentration might have been caused by immobilization of the N in the soil, a problem with cherry tomato diseases, and interspecific competition between a cherry tomato and green manure.

Corn straw with a high C:N ratio under the soil can cause immobilization of N, as previously reported by [31–34]. Additionally, the C:N ratio of the soil increased between the cultivation. The repetition of the cherry tomato crop increases the problem with diseases such as late blight (*Phytophthora infestans*), which affects the leaves of cherry tomatoes and consequently affects their photosynthetic capacity and reduces N absorption [22,42]. The cherry tomato was debilitated because of the disease in 2012 that may have facilitated competition between the legume and the cherry tomato for space, water, and nutrients, although the legume produced a similar dry mass in 2011/12 in general (8.3 and 73 Mg ha$^{-1}$).

The leaves and fruits received more N transfer than the shoots (Figure 4). It was shown that the N transfer increases with the growth/development of the cherry tomato. This observation can be likely associated with the N supply's increasing necessity for a portion of cherry tomato. Furthermore, as the legume grows and develops, it also increases the N accumulation from BNF, the relationship between the roots (legume and non-legume), and the interaction between mycorrhizae and roots, consequently increasing the possibility of N transfer to a cherry tomato.

Although the N concentration of cherry tomato was higher in 2012 than in 2011, the yield was 41% lower in 2012 (864 g plant$^{-1}$ of fruits) than in 2011 (1475 g plant$^{-1}$). The cherry tomato showed less growth in 2012, and this high N concentration is probably the result of the concentration of this nutrient in less dry mass. However, we did not observe symptoms of N deficiency independent of the year, and thus we concluded that the soil and the legumes provided sufficient N for the cherry tomato.

This study did not show a difference in N transfer ($p > 0.10$) for the treatments (Figure 4). Thus, the N of legumes was transferred to cherry tomatoes in similar quantities. However, the absence of the capacity of N transfer of legumes to non-legumes might have been masked for one possible N contamination between the plots. As discussed in the study of [22], the plot proximity possibly allowed cross-contamination of N between them, because N is highly mobile and can be transferred between plants for many pathways and over long-distances (>1.5 m), such as ammonia gas leaves, root exudates, and mycorrhizae [14,17,43]. In future studies with nitrogen transfer in the field, it will be essential to take care to randomize the plot and use greater distances between the plots or even barriers to avoid cross-contamination.

The green corn in 2013 had less N concentration in the leaves and aboveground biomass and less N accumulation in the aboveground biomass (Table 9). Despite the cultivation of legumes in the intercropping system before green corn, the absence of specific fertilization for green corn might have caused a reduction of the N available in the soil and consequently reduced the absorption by green corn. Furthermore, the N concentration of the leaves was lower than recommended for the corn [44–46], and the N accumulated in the aboveground biomass was less than that found by [7,47]. The legume cultivated in the succession of green corn did not provide sufficient N to supply the green corn demand.

The intercropping system with legumes did not affect the $^{15}$N natural abundance of the leaves and aboveground biomass of green corn cultivated in succession (Figure 5). The leaves in 2013 had less than $^{15}$N natural abundance ($\delta^{15}$N) (Figure 5). The leaves reflect the $\delta^{15}$N of N available in the soil, and, consequently, the $\delta^{15}$N of the plant can be altered due

to mycorrhizae dependence, the N forms absorbed, and the depth of acquisition within the soil profile [48]. Therefore, the reduction in N available in the soil for green corn may have increased mycorrhizae dependence. The plants infected by mycorrhizae displayed $\delta^{15}N$ reduced by 2 ‰ compared to non-infected plants [48,49].

## 5. Conclusions

The BNF was responsible for more than half of the N accumulated in the legumes. The N of legumes was transferred to cherry tomato in similar quantities, and the leaves and fruits of cherry tomato received more N transfer than shoots. It was shown that N transfer increases with the growth/development of cherry tomatoes. The intercropping system with legumes did not affect the $^{15}N$ natural abundance of leaves and the aboveground biomass of green corn cultivated in succession. The legume in an intercropping system with cherry tomatoes cultivated in green corn succession does not provide sufficient nitrogen to supply the green corn demand.

**Author Contributions:** Conceptualization, E.J.A. and F.R.; methodology, E.J.A., F.R., and G.C.S.; Formal analysis, G.C.S., C.A.S., and T.M.; data curation, G.C.S.; statistic, I.P.O. and G.M.B.A.; writing—original draft preparation, G.C.S.; writing—review and editing, P.C.O.T. and E.J.A. All authors have read and agreed to the published version of the manuscript.

**Funding:** This research received no external funding.

**Institutional Review Board Statement:** Not applicable.

**Informed Consent Statement:** Not applicable.

**Acknowledgments:** The authors acknowledge São Paulo Research Foundation (FAPESP) scholarships granted to the Gabriela Cristina Salgado (2014/22602-5; 2018/25483-8), Edmilson Jose Ambrosano (2010/07666-6), and Fabrício Rossi (2011/05648-3), the Brazilian National Council for Scientific and Technological Development (CNPq) (142176/2018-4) scholarship granted to the Gabriela Cristina Salgado, and technical support on the research of APTA and Piraí seeds. This study was financed in part by the Coordenação de Aperfeiçoamento de Pessoal de Nível Superior—Brasil (CAPES)—Finance Code 001.

**Conflicts of Interest:** The authors declare no conflict of interest.

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
