# Peer review of "Biological N Fixation and N Transfer in an Intercropping System between Legumes and Organic Cherry Tomatoes in Succession to Green Corn"

_agriculture, doi:10.3390/agriculture11080690_

Round 1

Reviewer 1 Report

Summary:

This research paper, by Salgado et al., studies the biological N fixation and N transfer in an intercropping system between legumes and organic cherry tomatoes .

Overall, the topic is of interest to Agriculture (MDPI). However, there are some minor issues the way they presented their results and how they interpreted the results in the discussion section.

I suggest some revisions for this manuscript.  I have clearly indicated all my opinions and suggestions below.

Broad Comments:

1.The methods section fails to describe the N natural abundance procedure employed in the study. This is key to this study and needs to be clearly stated.

  1. The discussion section ends abruptly and a clear “conclusion section” to list out the salient features of the study would help the flow.

Specific Comments

Table 1.  Please follow standard metric units for soil characteristics. PPM or mg/kg

Line 47 and 56:  please change “culture” to maybe crop or plants.

Line 102: What do you mean by “area experimental”

Line 369: How would you defend this sentence regarding randomization?  You already  had a CRBD design.

Author Response

Dear Reviewer, 

Thank you for your comments.

  1. The 15N natural abundance is one method used to calculate the N transfer and the biological nitrogen fixation. I described it briefly because this method is very used for these calculations. However, I related more details to topic 2.4, page 20.
  2. I included the Conclusion Topic.
  3. I accepted your comments.
  4. In relation the line 369, I didn't understand your comment. Can you rewrite it for me, please?

Reviewer 2 Report

The topic is well known and of interest. The results are about 10 years old, why this paper is published late?

Why do you used the level of significance for analyses of variance  p<0.10 and not p<0.05 ? Why different levels of significance are used in table 3 and 6 (and described in line 252?

Cited literature  is ok.

The main question is about the biological N fixation and N transfer in an intercropping system, which is the standard in agricultural practice. The topic is new for the use of organic cherry tomatoes. Thus the research is using a standard method and procedures as well. For open field research it would be better to have results of 3 years because the growing conditions are changing from year to year. It was enough time, because the results are from 2011-2013.

The topic itself is original and completed the knowledge about the biological N fixation from different legumes and transfer to the cherry tomatoes.

Compared with other publishers the research and results are similar but for other main crops like tomatoes. Furthermore the results are not a scientific surprise, but following the publishers before.

The paper itself is well written and clear and easy to read. The conclusion represents all the results with the necessary evidence and arguments.

Author Response

Dear Reviewer,

Thank you for your comments.

1. The work was not published before because the analysis cost was very expensive, and we needed to approve projects to be able to pay for them. Despite completing 10 years of experiment, it remains current and relevant to research. This is because very few works with N transfer and biological N fixation with horticultural plants and in the field. In addition, this work can provide the basis for future research on nitrogen transfer using the 15N natural abundance method and for plant nutrition in organic agriculture. The organic farm has grown in the last century, driven by concern for the environment and safe food. Among the nutrients required by crops, nitrogen is the most limiting factor for high crop yield; thus, sustainable agricultural practices capable of supplying nitrogen are essential for organic production. One way to supply nitrogen sustainably is through the transfer of N between leguminous and non-leguminous platforms in the intercropping system. 

2. Statistical analysis was carried out with 10%. We understand that in a field experiment, the level of significance is chosen by risk. Using a 10% significance level means that if you repeat the experiment, you have a 90% probability of repeating the result, relevant to agronomy, especially in the field experiment, which does not occur in medicine, which requires 1 to 5% of significance. In addition, we tested ourselves for the 5% probability level as well. However, the differences seen by the high difference between means and difference in the field showed no difference at 5%. Following are the examples of studies that publication with 10% of significance: https://doi.org/10.1080/01904167.2020.1724304;https://doi.org/10.4025/actasciagron.v40i1.36530;https://doi.org/10.1590/S0103-90162011000300014; https://doi.org/10.4025/actasciagron.v33i4.6766; https://doi.org/10.1590/S0103-90162013000500005.

3. Tables 3 and 6 used the 1 or 5% significance level for the Dunnet test only. This test compares the 15N natural abundance of each treatment with the 15N natural abundance of Non-legume (table 3) or Reference plants (table 4). A statistical difference between the treatments and Non-legume or Reference plants means that they have Biological Nitrogen fixation or N transfer, respectively. Additionally, this method is based on a slight δ15N ‰ difference between N-fixation and other N sources. Thus, the level of the statistics needs to be more rigorous.

4. I included the Conclusion Topic. 

Reviewer 3 Report

Comment The submitted work for the review „Biological N fixation and N transfer in an intercropping system between legumes and organic cherry tomatoes in succession to green corn”, meets the requirements in the journal Agriculture, taking into account the comments included in the text. The article is written well, however the value of the article is diminished by the fact that the presented data comes from almost 10 years ago.

Author Response

Dear Reviewer, 

Thank you for your comments, 

Table 5 did not include the N accumulated derived of biological nitrogen fixation (Ndf) of white lupine in 2012 because there was not BNF this year.

The other comments in the text have been corrected in the text.

The work was not published before because the cost of the analysis was very expensive and we needed to approve projects in order to be able to pay for them.Despite completing 10 years of experiment, it remains current and relevant to research. This is because there are very few works with N transfer and biological N fixation with horticultural plants and in the field. In addition, this work can provide the basis for future research on nitrogen transfer using the 15N natural abundance method and for plant nutrition in organic agriculture. The organic farm has grown in the last century, driven by concern for the environment and safe food. Among the nutrients required by crops, nitrogen is the most limiting factor for high crop yield; thus, sustainable agricultural practices capable of supplying nitrogen are essential for organic production. One way to supply nitrogen sustainably is through the transfer of N between leguminous and non-leguminous platforms in the intercropping system.